# *Lactobacillus Mucosae* Strain Promoted by a High-Fiber Diet in Genetic Obese Child Alleviates Lipid Metabolism and Modifies Gut Microbiota in *ApoE^-/-^* Mice on a Western Diet

**DOI:** 10.3390/microorganisms8081225

**Published:** 2020-08-12

**Authors:** Tianyi Jiang, Huan Wu, Xin Yang, Yue Li, Ziyi Zhang, Feng Chen, Liping Zhao, Chenhong Zhang

**Affiliations:** 1State Key Laboratory of Microbial Metabolism, School of Life Sciences and Biotechnology, Shanghai Jiao Tong University, Shanghai 200240, China; jiangtianyi@sjtu.edu.cn (T.J.); wuhuan1024@126.com (H.W.); 6yangxin6@163.com (X.Y.); liyue--230@sjtu.edu.cn (Y.L.); zhangzy1107@sjtu.edu.cn (Z.Z.); cf2001@sjtu.edu.cn (F.C.); lpzhao@sjtu.edu.cn (L.Z.); 2Department of Biochemistry and Microbiology and New Jersey Institute for Food, Nutrition and Health, School of Environmental and Biological Sciences, Rutgers University, New Brunswick, NJ 08901, USA

**Keywords:** *lactobacillus mucosae*, human-derived probiotics, next generation probiotics, gut microbiota, lipid metabolism, atherosclerosis, *ApoE^-/-^* mice

## Abstract

Supplementation of probiotics is a promising gut microbiota-targeted therapeutic method for hyperlipidemia and atherosclerosis. However, the selection of probiotic candidate strains is still empirical. Here, we obtained a human-derived strain, *Lactobacillus mucosae* A1, which was shown by metagenomic analysis to be promoted by a high-fiber diet and associated with the amelioration of host hyperlipidemia, and validated its effect on treating hyperlipidemia and atherosclerosis as well as changing structure of gut microbiota in *ApoE^-/-^* mice on a Western diet. *L. mucosae* A1 attenuated the severe lipid accumulation in serum, liver and aortic sinus of *ApoE^-/-^* mice on a Western diet, while it also reduced the serum lipopolysaccharide-binding protein content of mice, reflecting the improved metabolic endotoxemia. In addition, *L. mucosae* A1 shifted the gut microbiota structure of *ApoE^-/-^* mice on a Western diet, including recovering a few members of gut microbiota enhanced by the Western diet. This study not only suggests the potential of *L. mucosae* A1 to be a probiotic in the treatment of hyperlipidemia and atherosclerosis, but also highlights the advantage of such function-based rather than taxonomy-based strategies for the selection of candidate strains for the next generation probiotics.

## 1. Introduction

Hyperlipidemia, which results from an imbalanced lipid metabolism, is considered as a strong risk issue for the development of atherosclerosis [1,2]. The elevations of cholesterol or/and triglyceride in plasma may be related to multiple factors, including dietary pattern, lifestyle (e.g., physical activity, smoking and alcohol consumption), physiopathology (e.g., age, gender, insulin resistance and central obesity), and genetics [3]. Because of the multifarious pathologic process, the underlying mechanism of hyperlipidemia remains unclear [4].

Gut microbiota, a complex ecosystem consisting of trillions of microorganisms, is deeply involved in the dyslipidemia and atherosclerotic development of the host [5]. Compared with germ-free (GF) mice, the level of triglyceride of conventionally raised (CONV-R) mice reduced in plasma but increased in adipose tissue and liver [6,7]. GF mice were resistant to diet-induced obesity with increased fatty acid metabolism [8]. In addition, human studies revealed that specific taxa as well as the richness of gut microbiota were associated with the level of lipid in the plasma of atherosclerotic patients [9]. Studies have revealed several mechanisms of gut microbiota participating in the lipid metabolism of the host, such as the metabolism of bile acids [10], the production of short chain fatty acids (SCFAs) [11], and the translocation of lipopolysaccharide (LPS) over the gut barrier [12]. Plasma trimethylamine-*N*-oxide (TMAO) is a new identified risk factor for hyperlipidemic atherosclerosis [13]. Dietary choline is metabolized firstly to trimethylamine (TMA) by gut microbiota, then to TMAO by enzymes of the flavin monooxygenase (FMO) family in the liver [13]. Gut microbiota can modulate the generation of TMAO not only through the pathway for TMA production [13], but also through altering FMO3 (the most active member in metabolizing TMA to TMAO of FMO family in the liver) expression via processing bile acids [14]. Thus, gut microbiota may serve a promising therapeutic target for managing hyperlipidemia and atherosclerosis [15].

Supplementation of probiotics is an effective method to correct disturbed gut microbiota and may alleviate excessive lipid accumulation and protect against atherosclerosis [15,16]. However, the rational selection of candidate strains is critical for developing effective probiotics. Different strains in the same species but isolated from different sources may exhibit different functions on the lipid metabolism of the host. For example, L3 and L10 were two strains from a well-characterized probiotic bacterial species *Lactobacillus reuteri*, where only L3 isolated from normal mice was shown to be anti-inflammatory in vitro and L10 isolated from obese mice failed to do so [17]. Rather than relying on taxonomy, more functional data are needed for guiding the selection of probiotic candidates [18]. We need to integrate tools such as metagenomics for more efficient selection and isolation of probiotic candidate strains. 

In our previous clinical study, we used a high-fiber dietary formula to modulate the gut microbiota which contributed to alleviation of lipid metabolism in the obese children with Prader-Willi syndrome (PWS) [19]. Metagenomics analysis based on high quality draft genomes showed that among the altered members in the gut microbial community of one child from that cohort under dietary intervention, the genome of *Lactobacillus mucosae* was dramatically increased and may contribute to the reduction of triglyceride in the plasma of the host [19,20]. In this study, we isolated a strain A1 of *L. mucosae* from the post-intervention fecal sample of the PWS child. In *ApoE^-/-^* mice on a Western diet, an animal model of hyperlipidemia and atherosclerosis [21], we showed that this strain alleviated severe lipid accumulation and atherosclerosis, in conjunction with the shifted the gut microbial structure. This study not only verified the causative contribution of *Lactobacillus mucosae* to the alleviation of host lipid metabolism, but also suggests that *L. mucosae* A1 can be a promising candidate probiotic in the treatment of hyperlipidemia and atherosclerosis. Thus, integrating metagenomics with the guided isolation of candidate strains may lead to more promising probiotics.

## 2. Materials and Methods

### 2.1. Sequence-Guided Isolation and Identification of A1 strain

We used a “sequence-guided isolation” scheme to isolate the predominant *L. mucosae* from the gut of the PWS child after the child received a 105-day high-fiber dietary intervention. The total bacterial DNA of the child’s fecal samples collected at several time points during the intervention was extracted. The V3 regions in 16S rRNA gene of these DNA samples were amplified with universal bacterial primers P2 (5′-ATTACCGCGGCTGCTGG-3′) and P3 (5′-CGCCCGCCGCGCGCGGCGGGCGGGGCGGGGGCACGGGGGGCCTACGGGAGGCAGCAG-3′), and analyzed by denaturing gradient gel electrophoresis (DGGE) [22]. In the DGGE fingerprinting of the V3 regions in the 16S rRNA gene from the fecal samples, bands of HA1, HA7 and HA12 became obvious after 105 days of dietary intervention compared to before the intervention (Appendix A). The HA1 band shared 100% sequence similarity with the V3 region in the 16S rRNA gene of *Lactobacillus acidophilus* 30SC; the HA7 band shared 100% sequence similarity with the V3 region in the 16S rRNA gene of *Lactobacillus mucosae* S32; and the HA12 band shared 100% sequence similarity with the V3 region in the 16S rRNA gene of *Bifidobacterium pseudocatenulatum* B1279. An amount of 0.6 g frozen 105-days fecal sample was suspended in 30 mL anaerobic Ringer’s solution (containing 0.1% *L*- cysteine) in an anaerobic chamber (A45; Don Whitley Scientific Ltd., Shipley, UK). Fecal matter was serially diluted tenfold with Ringer’s solution (containing 0.1% *L*- cysteine). Then, 200 μL of the 10^−5^ dilution was spread onto de Man, Rogosa, and Sharp (MRS) broth (HB0384-1; Qingdao Hope Bio-Technology Ltd., Qingdao, China) ager plates (containing 0.05% *L*- cysteine) and incubated at 37 °C under anaerobic conditions for 18 h. In total, 168 isolates were selected randomly and purified by streaking two times on MRS broth ager plates. In the DGGE profiles of V3 regions in 16S rRNA gene of 168 isolates, 88 isolates migrated to the identical position as HA7 band (Appendix A, DGGE I and II types). Enterobacterial Repetitive Intergenic Consensus (ERIC) sequences of the genome from these 88 isolates were amplified by polymerase chain reaction (PCR) with primers ERIC1 (5′-ATGTAAGCTCCTGGGGATTCAC-3′) and ERIC2 (5′-AAGTAAGTGACTGGGGTGAGCG-3′) [23]. The ERIC-PCR profiles classified 88 isolates into 4 types (Appendix A, E1–E4), and isolates sharing the same ERIC-PCR pattern were considered as the same strain. The A1 strain was randomly selected as the representative of E1, which had the highest number of isolates, and considered as the representative of the predominant *L. mucosae* in the gut of the PWS child after 105-days dietary intervention. The A1 strain was further identified by 16S rRNA gene sequencing and used in the animal trial.

### 2.2. Whole-Genome Sequencing and Analysis of A1 Strain

Whole-genome DNA of A1 strain was extracted with the blood and cell culture DNA kit (13323; Qiagen Inc., Valencia, CA, USA) and sequenced with the GridION platform (Oxford Nanopore Technologies Ltd., Oxford, UK). Raw reads were filtered with mean_qscore_template >= 7 and sequence_length >= 1000 bp. The filtered reads were assembled by Canu v1.7.11 under default setting [24]. Then Canu assemble were polished by Pilon v1.22 with second-generation sequencing data under default setting [25]. After removing the redundance, Canu assemble were corrected by Circulator v1.5.5, in which the -fixstart option was applied to set the *dnaA* gene as the first gene [26]. The genome sequence of *L. mucosae* LM1 was downloaded from the national center for biotechnology information (NCBI) with accession no. SAMN02470226. 

The average nucleotide identity (ANI) value was calculated between the genomes of A1 strain and other *L. mucosae* strains using ANI calculator available online [27]. Protein-coding sequences (CDSs), tRNAs, and rRNAs were predicted and annotated using the Prokka v1.13.3 pipeline [28]. Predicted CDSs were assigned to different categories in the clusters of orthologous groups (COG) using eggnog-mapper [29] based on eggnog 4.5 orthology data [30]. Genes related in carbohydrate utilization were annotated using dbCAN2 meta server [31]. The identifications of exopolysaccharide (EPS) gene cluster were performed by searching those highly conserved genes reported in the characterized EPS gene clusters in *L. mucosae*, followed by manually checking the annotation of the adjoining genes as well as subjecting them to BLAST [32]. Potential bacteriocin operons were predicted using BAGEL4 [33].

### 2.3. Culture and Administration of A1 Strain

A1 strain was cultured anaerobically (80% N_2_, 10% CO_2_, and 10% H_2_) in MRS broth at 37 °C. For bacteria for gavage, A1 strain was cultured in 400 mL volume of MRS broth medium until its stationary phase. Bacteria were collected after centrifugation, washed once with 40 mL anaerobic phosphate buffered saline (PBS, pH = 7.4), then resuspended in 2 mL anaerobic PBS to a density of 5 × 10^10^ cfu/mL. The suspension was distributed into 5 mL centrifuge tubes (200 µL/tube), added with 200 µL PBS containing 50% glycerin, mixed thoroughly and stored at −80 °C. Before gavage, the stored bacterial cells were defrosted and diluted with 3600 µL of anaerobic PBS.

### 2.4. Animal Trial and Sample Collection

All the animal experimental procedures were approved by the Institutional Animal Care and Use Committee of the School of Life Sciences and Biotechnology, Shanghai Jiao Tong University (approval ID: No. A2018045, approval date: 14/6/2018). An amount of 24 male specific-pathogen-free *ApoE^-/-^* mice (8-to-10-week-old) with C57BL/6 background were caged 2–4 per cage and randomly assigned to three groups: (1) NC group (*n* = 7), in which the mice were fed with a normal chow diet (1010009; 4.8% fat; Jiangsu Xietong Pharmaceutical Bio-engineering Ltd., Nanjing, China); (2) WD group (*n* = 8), in which the mice were fed with a high-fat high-cholesterol Western diet (D12079B; 21% fat and 0.15% cholesterol; FBSH Biotechnology Ltd., Shanghai, China); (3) WD+LM group (*n* = 9), in which the mice were fed with the Western diet and daily gavaged with 1 × 10^9^ cfu *L. mucosae* A1 suspended in 200 µL anaerobic PBS containing 2.5% glycerin. The mice in NC and WD group were daily gavaged with 200 µL anaerobic PBS containing 2.5% glycerin as a control. The treatment lasted for 13 weeks. Body weight of mice was measured once a week. Food intake per cage was measured once every two days. Fecal samples of all the mice were collected at the baseline, 4th, 8th and 13th week, then stored at −80 °C until gut microbiota profiling. At the 4th and 8th week of intervention, blood samples were collected from by tail bleeding after 6-h fasting. In detail, we secured mice in a restraint tube, wiped the tail with warm water to cause vasodilation, snipped the very end of the tail with a sterile number 11 scalpel, stroked the tail to collect blood into a collection tubes and stopped bleeding by applying pressure with a gauze pad as well as styptic powder to the tail tip. Serum samples were isolated by centrifugation at 3000 g at 4 °C for 15 min and stored at −80 °C until lipid, TMA and TMAO measurements following the procedures described below. At the end of the experiment, all the mice were anesthetized with isoflurance after 6-h fasting. Blood samples were collected by retro orbital bleeding. After exsanguination, mice were killed by cervical dislocation. Serum samples were isolated and stored at −80 °C until lipid, TMA, TMAO and lipopolysaccharide-binding protein (LBP) measurements. Liver, adipose tissues (epididymal, mesenteric, subcutaneous and retroperitoneal) and heart with aortic sinus were collected, weighed and stored at −80 °C or 4% paraformaldehyde until the following analyses.

### 2.5. Histopathology of Epididymal Adipose Tissue (eAT), Liver and Aortic Sinus

For eAT and liver, fresh tissue were fixed with 4% paraformaldehyde for at least 48 h, embedded in paraffin, sectioned, and stained with hematoxylin and eosin (H&E) (G1003; Wuhan servicebio technology Ltd., Wuhan, China). Adipocyte size and liver steatosis score were assessed using Image Pro Plus v6.0 (Media Cybernetics Inc., Silver Springs, MD, USA). For each mouse, mean areas of adipocytes were determined in at least five discontinuous scans under ×100 magnification, and scores of liver steatosis were assessed as described previously [34].

For analysis of atherosclerotic lesion in aortic sinus, the proximal aorta attached to heart was harvested, fixed in 4% paraformaldehyde for at least 48 h, embedded in optimal cutting temperature (OCT) compound (4583; CellPath Ltd., Newtown, Powys, UK) and frozen at −20 °C. From the beginning of aortic root to the extent for 200 μm, four sections (8-μm thickness) taken at 50 μm intervals were collected from each mouse. Sections were stained with Oil Red O (O0625; Sigma-Aldrich Ltd., St. Louis, MO, USA). The area of atherosclerotic lesion in the aortic sinus was measured using Image Pro Plus v6.0 and expressed as the mean size of the four sections for each mouse.

### 2.6. Hepatic and Serum Lipid Measurement

Assay kits were used to measure concentrations of serum total cholesterol (A111-1-1; NanJing Jiancheng Bioengineering Institute, Nanjing, China) and triglyceride (A110-1-1; NanJing Jiancheng Bioengineering Institute, Nanjing, China). Frozen liver sample was homogenized in a corresponding volume (W/V:1/9) of homogenizing buffer (pH 7.4, 0.01 mol/L Tris-HCl, 0.1 mmol/L EDTA-2Na, 0.8% NaCl). The supernatant was collected after being centrifuged at 2000 g for 25 min at 4 °C and used to measure concentrations of hepatic total cholesterol and triglyceride using the same kits described above. The results of hepatic lipid were corrected by total protein concentration (20202ES76; Yeasen Biotechnology Ltd., Shanghai, China). 

### 2.7. Serum TMA and TMAO Measurement

An amount of 20 μL of serum was added with 80 μL of acetonitrile and then placed at −20 °C for 30 min to precipitate proteins. Samples were centrifuged at 14,000 rpm for 15 min at 4 °C. Then, 25 μL of supernatant was collected and derivatized with 75 μL of 50 mM *tert*-butyl bromoacetate (124230; Sigma-Aldrich Ltd., St. Louis, MO, USA) in acetonitrile and 10 μL of 70% ammonium hydroxide in water. After standing at room temperature for 30 min, each sample was added with 1 mL of 50% acetonitrile/50% water containing 1% formic acid. Samples were centrifuged at 14,000 rpm for 5 min at room temperature, and the supernatants were analyzed by liquid chromatograph-mass spectrometer (LC/MS). LC/MS analysis was performed using a 2D H-Class ACQUITY UPLC coupled with Xevo™ TQ-S Series Triple Quadrupole mass spectrometer (2D H-class & TQ-XS) (WATERS Inc., Milford, MA, USA). Chromatographic separations were performed on an ACQUITY UPLC BEH HILIC Column (50 × 2.1 mm, internal diameter of 1.7 μm; WATERS Inc., Milford, MA, USA) and protected by ACQUITY UPLC BEH HILIC VanGuard^TM^ Pre-column (2.1 × 5 mm, internal diameter of 1.7 μm; WATERS Inc., Milford, MA, USA). The column was heated to 40 °C, and the flow rate was maintained at 0.5 mL/min. The gradient for elution was 5% A for 1 min, to 20% A in 4.5 min, to 60% A in 5 min, at 60% A for 1.5 min, to 5% A in 6.7 min, at 5% A for 1.8 min, where A was water with 10 mM ammonium formate and B was 95% acetonitrile/5% water with 10 mM ammonium formate. The mass spectrometer was equipped with an electrospray ionization source. The capillary voltage was set at +3000 V and heated to 550 °C. TMA and TMAO were monitored in multiple reaction monitoring mode using characteristic precursor-product ion transitions: m/z 174→118 for TMA, and m/z 76→58 for TMAO.

TMA (T72761; Sigma-Aldrich, Ltd., St. Louis, MO, USA) and TMAO (317594; Sigma-Aldrich, Ltd., St. Louis, MO, USA) were used to make standards at various concentrations and followed by the procedures for derivatization and LC/MS analysis above. The levels of TMA and TMAO in serum were quantified with standard curve using peak area under the MassLynx v4.2 (WATERS Inc., Milford, MA, USA). 

### 2.8. Serum LBP Measurement

Enzyme-linked immunosorbent assay kits (PCDBM0177; Shanghai P&C Biotechnology Ltd., Shanghai, China) were used to determine the concentration of serum LBP according to the manufacturer’s instructions.

### 2.9. Statistical Analysis for Animal Trial

Statistical significances for physiological and biochemical data of mice among different groups were analyzed by one-way analysis of variance (ANOVA) test followed by a Tukey post hoc test using “stats” R package in R v3.6.2 (www.r-project.org). Differences were considered significant when *p* < 0.05.

### 2.10. Gut Microbiota Profiling

The bacterial DNA extraction from fecal samples of *ApoE^-/-^* mice collected at the baseline, 4th, 8th and 13th week was performed as previously described [35]. A sequencing library of the V3-V4 region of the 16S rRNA gene was constructed following the manufacturer’s instructions (Part # 15044223Rev.B; Illumina Inc., San Diego, CA, USA) with improvement as previously described [36], and sequenced on the Illumina MiSeq platform (Illumina, Inc., San Diego, CA, USA).

The raw paired-end reads were submitted to Quantitative Insights Into Microbial Ecology2 (QIIME2) V2018.11 for analysis [37]. Adapters and primers were removed with q2-cutadapt [38]. Plugin “qiime dada2 denoise-paired” in QIIME2 was then used for quality trimming, denoising, merging and chimera detection with default setting except for “-p-trunc-len-f” and “-p-trunc-len-r” which were set at 269 and 179, respectively [39]. All amplicon sequence variants (ASVs) were aligned with mafft and built into a rooted phylogenetic tree with fasttree2 via q2-phylogeny [40,41]. ASVs were taxonomically classified using q2-feature-classifier [42]. Alpha diversity metrics (observed ASVs and Shannon index), beta diversity metrics (weighted UniFrac distance [43]), and principle coordinate analysis (PCoA) were calculated using q2-diversity after samples were rarefied to 22,000 sequences per sample. The statistical significances of observed ASVs and Shannon index was analyzed by one-way ANOVA test followed by a Tukey post hoc test using “stats” R package in R v3.6.2. The statistical significances of the weighted UniFrac distance of gut microbiota from WD group to NC group against that from WD+LM group to NC group were analyzed by unpaired *t*-test (two-tailed) using “stats” R package in R v3.6.2. The statistical significance of gut microbiota structure among different groups was analyzed by permutational multivariate analysis of variance (PERMANOVA; 9999 permutations) via q2-diversity. Differences were considered significant when *p* < 0.05 or *Q* < 0.05.

Sparse partial least-squares discriminant analysis (sPLS-DA) models [44] were established to identify specific ASVs that contributed to the segregation of gut microbiota induced by the Western diet or *L. mucosae* A1 using “mixOmics” R package [45] in R v3.6.2. Centered log ratio (CLR) transformations were implemented in sPLS-DA to circumvent spurious results. The optimal classification performances of the sPLS-DA models were estimated by the perf function using leave-one-out (loo) cross-validation with the smallest balanced error rate. The heatmap showing the abundance variation of 47 selected ASVs in three groups was generated using “pheatmap” R package. Variations in the relative abundance of specific ASV among different groups were analyzed by Mann–Whitney U test (two-tailed) in MATLAB R2019b. Differences were considered significant when *p* < 0.05.

Correlations between ASV abundance and individual host parameters related to fat accumulation as well as atherosclerosis were identified using Spearman’s correlation in MATLAB R2019b. The *p*-values of these correlations were adjusted by false discovery rate (FDR) estimation introduced by Benjamini and Hochberg [46]. Correlations were considered significant when adjusted *p* < 0.05.

### 2.11. Accession Numbers for Sequence Data

The 16S rRNA gene sequence of *L. mucosae* A1 has been submitted to the GenBank with the accession no. MT742634. The complete genome sequence of *L. mucosae* A1 has been submitted to the NCBI with the accession no. SAMN15515560. The raw Illumina sequence data generated in this study have been submitted to the Sequence Read Archive (SRA) at NCBI under accession no. SRP271573.

## 3. Results

### 3.1. Isolation of Human-Derived L. Mucosae A1 Enriched by a High-Fiber Dietary Intervention

We isolated 88 isolates belonging to *L. mucosae* from the post-intervention fecal sample of the PWS child. Under a ‘sequence-guided isolation’ scheme, we obtained one isolate (named A1) as the representative of the *L. mucosae* enriched by the high-fiber dietary intervention (Appendix A). The cells of the A1 strain were (2.3−3.0) × (0.6−0.7) μm, rod-shaped, and with no flagella or other appendages (Figure 1A). After blasting against the 16S rRNA sequences database of bacteria and archaea on the NCBI, we found that the 16S rRNA gene of the A1 strain had the highest similarity with *L. mucosae* LM1 (99.87%) and the A1 strain belonged to the species *L. mucosae* (Figure 1B). The genome of the A1 strain consisted of a single circular chromosome and three detected plasmids (Appendix A). The ANI values between the genome of the A1 strain and all publicly available genomes of *L. mucosae* species on the NCBI were ranged from 97.22% to 96.12%, and the ANI value between the A1 strain and *L. mucosae* LM1 (the reference genome of *L. mucosae*) was 96.81%.

As a bacterium enriched by a high-fiber diet, the A1 strain harbored more genes involved in carbohydrate transport and metabolism compared to LM1 strain (8.17% in A1 vs. 6.88% in LM1) (Figure 1C; Appendix A). Compared to the LM1 strain, the A1 strain had more genes related to glycoside hydrolases (*n* = 27 in A1 vs. *n* = 22 in LM1) and less genes related to glycosyl transferase (*n* = 24 in A1 vs. *n* = 27 in LM1) (Appendix A). EPS produced by lactobacilli appears to contribute several benefits to their hosts, including competing against the attachment of pathogens, promoting the growth of beneficial bacteria and modulating the immune system [32]. The EPS gene clusters identified in A1 and LM1 were similar, except transpose genes existed differentially between the A1 strain (*n* = 2) and LM1 strain (*n* = 6) (Appendix A). Bacteriocin is another feature of lactobacilli, assisting the strains to compete within complex microbial communities and positively influence the health of the host [47,48]. There was one enterolysin A operon in the chromosome of both of these two strains, and only one plasmid of the A1 strain had another enterolysin A operon.

To summarize, we isolated the *Lactobacillus* sp., *L. mucosae* A1, which was enriched by the high-fiber dietary intervention, and this human-derived strain could be used in further animal study to evaluate its effect in vitro.

### 3.2. L. mucosae A1 Alleviated Severe Lipid Accumulation in ApoE^-/-^ Mice on a Western Diet

Compared with normal chow-fed *ApoE^-/-^* mice, *ApoE^-/-^* mice on a Western diet exhibited a significant increase in body weight gain (Appendix A and Figure 2A), fat mass (Figure 2B) and adipocyte size (Figure 2C). Under daily supplementation with *L. mucosae* A1, the mice in WD+LM group showed lower body weight gain, fat mass and adipocyte size compared to the mice in WD group (Figure 2A–C), despite no reductions in energy intake (Appendix A). The Western diet also resulted in elevations in serum total cholesterol and triglyceride detected at the 4th, 8th and 13th week (Figure 2D). Supplementation with *L. mucosae* A1 did not show any influence on serum total cholesterol but can obviously attenuate serum triglyceride under the detection at the 4th and 8th week (Figure 2D). After the Western diet feeding for 13 weeks, *ApoE^-/-^* mice developed excessive lipid accumulation in the liver with increased liver weight (Figure 2E), promoted concentrations of liver total cholesterol and triglyceride (Figure 2F), as well as elevated volume density of liver steatosis (Figure 2G). Administration of *L. mucosae* A1 reduced the effects of the Western diet on liver, resulting in significantly lower weight, triglyceride content and steatosis of liver (Figure 2E–G).

Taken together, these results demonstrated the ability of *L. mucosae* A1 to mitigate abnormal lipid metabolism in Western diet-fed *ApoE^-/^*^-^ mice.

### 3.3. L. mucosae A1 Protected Western Diet-Fed ApoE^-/-^ Mice from Atherosclerosis

In the present study, the atherosclerotic plaque area was significantly larger in *ApoE^-/-^* mice fed with a Western diet than that in normal chow-fed *ApoE^-/-^* mice, and *L. mucosae* A1 treatment reversed this trend of atherosclerosis (Figure 3A). Serum LBP of the mice in WD group was significantly higher than that of the mice in NC group reflecting a higher level of endotoxemia induced by the Western diet, which was lessened by *L. mucosae* A1 treatment (Figure 3B). However, the Western diet feeding severely lowered the level of serum TMA and TMAO, on which *L. mucosae* A1 treatment did not show any influence (Figure 3C). Considered together, the function of *L. mucosae* A1 on impeding atherosclerotic development of *ApoE^-/-^* mice may be associated with reduced metabolic endotoxemia-induced inflammation, but not related to the metabolism of TMA or TMAO.

### 3.4. L. mucosae A1 Modulated the Gut Microbiota in ApoE^-/-^ Mice on a Western Diet

To determine how gut microbiota of *ApoE^-/-^* mice was affected by the Western diet and *L. mucosae* A1 treatment, sequencing of microbial 16S rRNA gene V3-V4 region was performed on the 96 fecal samples collected at the baseline (0th week) and the 4th, 8th, 13th week of trial. After 13-weeks intervention, compared to the normal chow, the Western diet feeding decreased the richness and diversity of gut microbiota, which was reflected by a lower level of observed ASVs and Shannon index, but *L. mucosae* A1 treatment did not change this alteration of richness and diversity of gut microbiota by the Western diet (Figure 4A,B). PCoA and PERMANOVA of gut microbiota based on weighted UniFrac distance revealed a clear separation of microbial structure among the three groups (Figure 4C; Appendix A). The Western diet was the predominant factor in shaping gut microbiota, as the samples in two groups with the Western diet clearly separated from those in the NC group along the first principal component (PC1, accounting for 59.02% of total variance). *L. mucosae* A1 treatment also shaped gut microbiota structure, as samples in WD + LM group diverged from that in WD group along PC2 and PC4 (Figure 4C). These changes in richness, diversity and whole structure of gut microbiota induced by the Western diet and *L. mucosae* A1 were similar at the 4th and 8th week (Appendix A). Notably, the weighted UniFrac distance of gut microbiota in the mice between the WD + LM group and NC group emerged obviously higher than that between the WD group and NC group at the 13th week (Figure 4C). These results suggested that both the Western diet and *L. mucosae* A1 treatment changed the gut microbiota of *ApoE^-/-^* mice. 

The sPLS-DA models were constructed to identify members of gut microbiota whose abundance was changed by the Western diet (WD group vs. NC group, the balanced error rate for classification is 0, leave-one-out cross validation, Appendix A) or by *L. mucosae* A1 treatment (WD group vs. WD+LM group, the balanced error rate for classification is 0, leave-one-out cross validation, Appendix A) after 13-weeks intervention. As results, 260 ASVs and 47 ASVs were identified as the features of the two sPLS-DA models above, respectively. Compared to the normal chow, the Western diet enhanced the abundance of 137 ASVs and reduced the abundance of 123 ASVs. Supplementation of *L. mucosae* A1 reversed the changes of 29 ASVs that were altered by the Western diet, aggravated the accumulation of ASV159 that was induced by the Western diet and resulted in the changes of 17 ASVs that were not altered by the Western diet (Figure 4D; Appendix A). Thus, *L. mucosae* A1 not only can recover the disturbance of gut microbiota caused by the Western diet, but also can affect members of gut microbiota that were not related to the Western diet.

Spearman’s correlation analysis was performed between the abundances of 47 ASVs altered by *L. mucosae* A1 and specific host parameters related to lipid accumulation and atherosclerosis of the mice in all groups. In total, 29 ASVs were significantly correlated with at least one host parameter. Among these 29 ASVs, 28 ASVs showed positive correlation and one ASV (ASV236) showed negative correlation with disease phenotypes (Figure 4D). Supplementation of *L. mucosae* A1 reduced 26 of 28 ASVs that were positively correlated with disease phenotypes, including bacteria belonging to *Oscillibacter, Ruminiclostridium*, *Harryflintia*, *Enterorhabdus*, *Anaerovorax*, *Eubacterium*, *Turicibacter*, *Enterococcus*, unclassified Ruminococcaceae, unclassified Clostridiales, unclassified Lachnospiraceae (Figure 4D). These 26 ASVs were considered as key members of gut microbiota that potentially mediate the salutary effects of *L. mucosae* A1 on Western diet-induced lipid accumulation and atherosclerosis.

## 4. Discussion

In this study, we obtained a human-derived strain, *L. mucosae* A1, which was associated with the amelioration of host hyperlipidemia. In *ApoE^-/-^* mice fed with a Western diet, this strain reduced the accumulation of triglyceride in serum and liver, alleviated the development of atherosclerosis, and shifted the structure of gut microbiota. Then, *L. mucosae* A1 can be a new candidate of probiotics to alleviate hyperlipidemia and atherosclerosis.

In the current work, the selection of *L. mucosae* A1 from the gut of a host received a high-fiber dietary intervention was based on its association with the improvement of lipid metabolism, and the in vitro effect of this strain on hyperlipidemia was then validated in the animal model. It is viewed as a strategy for next generation probiotics selection [49]. The strategy overcomes the disadvantages in the selection of traditional probiotics, such as choosing microbes as probiotics dependent on its taxonomy without any knowledge of the effect of microbes on host health [49]. Similarly, *Akkermansia muciniphila* was chosen as it was decreased in obese mice, restored by prebiotic treatment, and significantly and inversely correlated with metabolic endotoxemia and related disorders [50]. Administration of *A. muciniphila* reversed obesity and gut barrier dysfunction in mice with a high-fat diet, appeared to be safe, and displayed the impact of improving several metabolic parameters on obese insulin-resistance individuals in a human study [50,51,52]. Under the strategy for next generation probiotics selection, we can efficiently find more potential probiotics targeting various gut-microbiota related diseases.

In our study, *L. mucosae* A1 did not modify the levels of TMA and TMAO in the plasma of the *ApoE^-/-^* mice on a Western diet. TMAO, which may promote atherosclerosis by inhibiting reverse cholesterol transport, has been recognized as a new target to protect against hyperlipidemic atherosclerosis [53]. However, not all gut microbiota targeted treatments for hyperlipidemia can influence the level of plasma TMAO [51]. Hyperlipidemic atherosclerosis is a complicated pathological process affected by multiple factors, except in the TMAO related pathway [54]. The chronic inflammation stimulated by the excessive penetration of LPS into circulation has been considered as one of the important factor [55], and the mechanism may be that LPS can activate Toll-like receptor 4 expressed in various vascular cells and perivascular adipose tissue, stimulate the release of proinflammatory cytokines, inhibit cholesterol efflux from macrophages, facilitate foam cell formation, and exacerbate atherosclerosis [55,56]. In our work, treatment with *L. mucosae* A1 reduced the LBP content in plasma, reflecting an attenuation of metabolic endotoxemia of the *ApoE^-/-^* mice on a Western diet. The decrease in chronic inflammation induced by bacterial antigen load might be the underlying pathway of *L. mucosae* A1 to attenuate hyperlipidemia and atherosclerosis in *ApoE^-/-^* mice. Then the molecular mechanism involved in it still needs further investigation.

Along with the improved metabolic endotoxemia and lipid metabolism, *L. mucosae* A1 influenced the structure of gut microbiota, a part of which is to recover the disturbance of gut microbiota caused by the Western diet. Among the members of gut microbiota altered by *L. mucosae* A1, the majority of microbes significantly positively related with host parameters for lipid accumulation and atherosclerosis were enriched by the Western diet but reduced by *L. mucosae* A1. These microbial cells may be involved in the development of metabolic endotoxemia and the deterioration of the lipid metabolism of *ApoE^-/-^* mice. For example, some of these bacteria were reported to participate in the intestinal barrier impairment of the host. *Oscillibacter* was found to be positively associated with increased permeability of mouse colon [57]. *Enterococcus* produces extracellular reactive oxygen species that damages colonic epithelial cells DNA [58]. In addition, some of these bacteria may promote the inflammation of the host. *Ruminiclostridium* was found to be positively correlated with serum LPS, TNF-α and IL-17A [59]. *Turicibacter* owns genes for internalin production and may have immunomodulatory or host invasion-related traits [60,61]. The amelioration of gastrointestinal microecology through reducing the level of these harmful gut microbes may be a mechanism for *L. mucosae* A1 to alleviate the metabolic endotoxemia and impaired lipid metabolism of *ApoE^-/-^* mice on the Western diet.

Upon the function of *L. mucosae* A1 on ameliorating lipid accumulation, the strain can not only be developed as a potential probiotic to treat hyperlipidemia and atherosclerosis, but may also be used together with specific dietary interventions, such as a ketogenic diet, to avoid side effects presented as elevated plasma lipid in treatment of other diseases [62,63,64]. However, the further preclinical studies are needed to reveal the mechanism of how *L. mucosae* A1 affects the metabolic endotoxemia and impaired lipid metabolism of the host with more animal models with hyperlipidemia [65], and the efficiency of *L. mucosae* A1 on alleviating severe lipid accumulation and shifting gut microbiota in patients with dyslipidemia and atherosclerosis requires to evaluated in clinical study.

Taken together, we obtained a human-derived potential probiotic, *L. mucosae* A1, which can alleviate severe lipid accumulation and shift the gut microbiota structure of *ApoE^-/-^* mice on a Western diet. These effects of *L. mucosae* A1 provide a rationale for the clinical development of the strain to prevent or treat dyslipidemia and atherosclerosis.

## Figures and Tables

**Figure 1 microorganisms-08-01225-f001:**
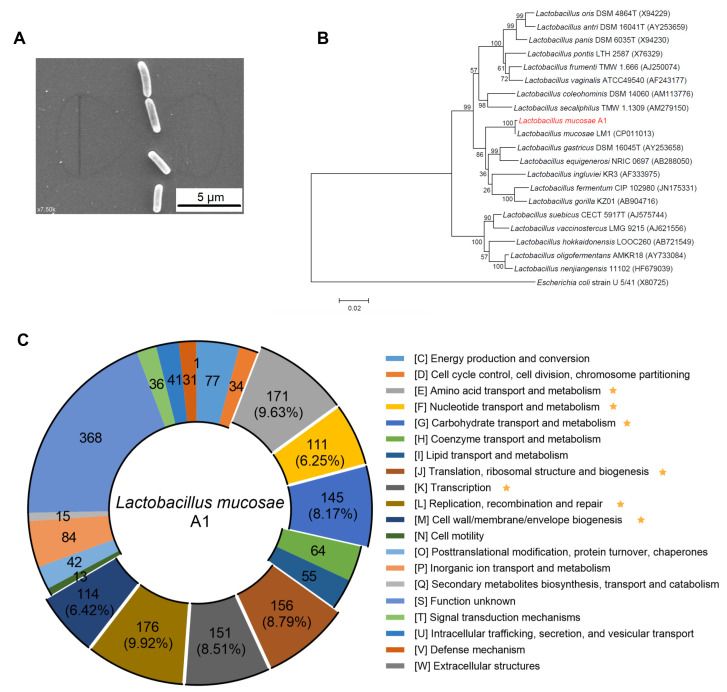
Identification and genomic characterization of *L. mucosae* A1. (**A**) Electron micrograph of A1 strain. The scale bar is 5 μm. (**B**) Phylogenetic relationships of the A1 strain with its relatives based on 16S rRNA gene sequences. The nearest neighbor of A1 strain was *L. mucosae* LM1. *Escherichia coli* U5/41 was used as an outgroup. The tree was constructed using the Neighbor-Joining method in MEGA6. The bar indicates sequence divergence. (**C**) Functional categories in the clusters of orthologous groups (COG) analysis of *L. mucosae* A1. The numbers out the brackets on the circle represents the numbers of genes in each COG category. The numbers in the brackets are the percentages of genes in the corresponding categories to the total genes identified by COG database. Those COGs have more than 5% of the total genes are marked with an asterisk.

**Figure 2 microorganisms-08-01225-f002:**
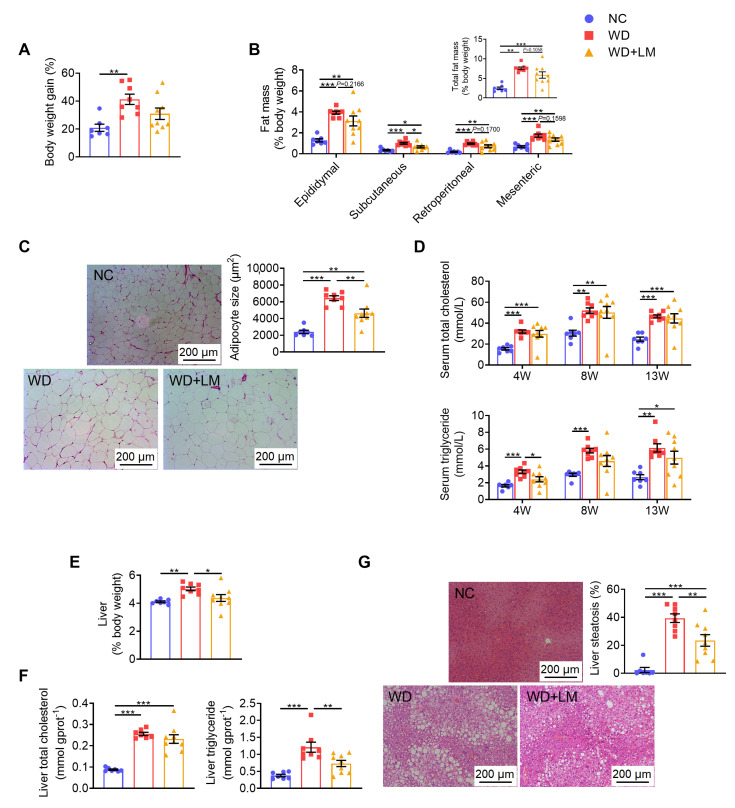
*L. mucosae* A1 alleviated lipid accumulation of *ApoE^-/-^* mice fed with a Western diet. (**A**) Body weight gain measured at the 13th week as the percentage of baseline weight for each mouse. (**B**) Epididymal, subcutaneous, retroperitoneal, mesenteric and total adipose tissue weight (ratio to body weight). (**C**) Representative photomicrographs of hematoxylin and eosin (H&E)-stained sections of epididymal adipose tissue (eAT) under 100× magnification and mean cell area of adipocyte in eAT. The scale bar is 200 μm. (**D**) The level of total cholesterol and triglyceride in serum collected at the 4th, 8th and 13th week. (**E**) Liver weight (ratio to body weight). (**F**) The level of total cholesterol and triglyceride in liver. (**G**) Representative photomicrographs of H&E-stained sections of liver under 100× magnification and volume density of liver steatosis. The scale bar is 200 μm. All the data are shown as means ± s.e.m. NC: *n* = 7, WD: *n* = 8, WD + LM: *n* = 9. Values of each group were analyzed by one-way analysis of variance (ANOVA) followed by Tukey’s post hoc test. * *p* < 0.05, ** *p* < 0.01, *** *p* < 0.001.

**Figure 3 microorganisms-08-01225-f003:**
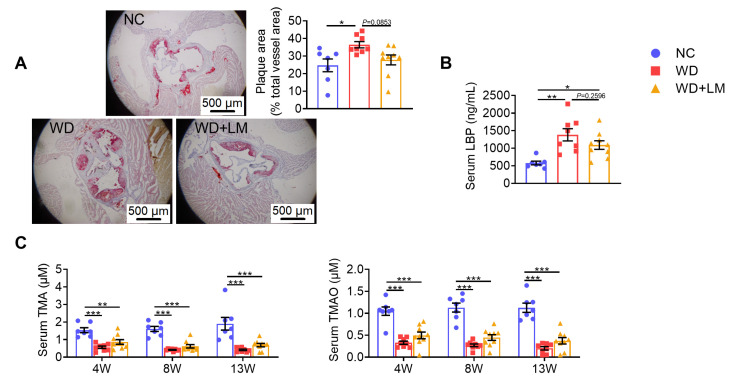
*L. mucosae* A1 protected Western diet-fed *ApoE^-/-^* mice from atherosclerosis. (**A**) Representative photomicrographs of Oil Red O-stained sections of aortic sinus under 40× magnification and area ratio of atherosclerotic plaque. The scale bar is 500 μm. (**B**) The level of lipopolysaccharide-binding protein (LBP) in serum. (**C**) The level of trimethylamine (TMA) and trimethylamine-*N*-oxide (TMAO) in serum collected at the 4th, 8th and 13th week. All the data are shown as means ± s.e.m. NC: *n* = 7, WD: *n* =8, WD + LM: *n* =9. Values of each group were analyzed by one-way analysis of variance (ANOVA) followed by Tukey’s post hoc test. * *p* < 0.05, ** *p* < 0.01, *** *p* < 0.001.

**Figure 4 microorganisms-08-01225-f004:**
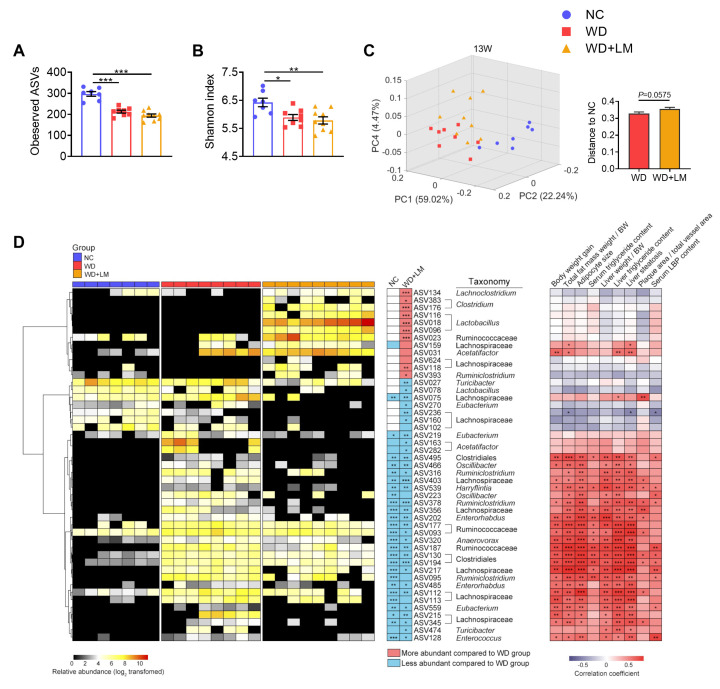
Modulation of gut microbiota in *ApoE^-/-^* mice after administration of a Western diet with or without *L. mucosae* A1 for 13 weeks. (**A**) Amplicon sequence variants (ASVs) richness of gut microbiota. (**B**) Shannon index of gut microbiota. (**C**) Principal coordinate analysis (PCoA) plot of gut microbiota based on weighted Unifrac distance and weighted Unifrac distance from WD group and WD + LM group to NC group. Data for ASVs richness, Shannon index and distance are shown as means ± s.e.m. Number of mice per group: NC: 7, WD: 8, WD + LM: 9. Values for ASVs richness and Shannon index of each group were analyzed by one-way analysis of variance (ANOVA) followed by Tukey’s post hoc test. Values for distance were analyzed by unpaired *t*-test. * *p* < 0.05, ** *p* < 0.01, *** *p* < 0.001. (**D**) 47 ASVs identified as key variables for differentiation between the gut microbiota of WD + LM group and that of WD group by sparse partial least squares discriminant analysis (sPLS-DA) model (shown in Appendix A). Left, the heatmap represents the normalized and log_2_-transformed relative abundance of the ASVs in each sample. The ASVs were clustered by the ward.D method. Middle, the changing direction of 47 ASVs in NC group and WD+LM group compared to WD group according to sPLS-DA models (shown in Appendix A). Redness and blueness indicate the relative abundance of ASVs were more and less, respectively, compared to WD group. The relative abundance of key ASVs were compared between groups by Mann–Whitney *U* test. * *p* < 0.05, ** *p* < 0.01, *** *p* < 0.001. Right, Spearman correlations between the host parameters related to lipid metabolism as well as atherosclerosis and the relative abundance of 47 ASVs. Colors red and blue denote positive and negative association, respectively. The intensity of the colors represents the degree of association between the abundances of ASVs and host parameters. *p* values of spearman correlations were adjusted by false discovery rate (FDR). * adjusted *p* < 0.05, ** adjusted *p* < 0.01, *** adjusted *p* < 0.001. The genus-level taxonomic classifications of the ASVs are shown. For all the figures, NC: *n* = 7, WD: *n* = 8, WD+LM: *n* = 9.

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
