# Peer review of "Lactobacillus Mucosae Strain Promoted by a High-Fiber Diet in Genetic Obese Child Alleviates Lipid Metabolism and Modifies Gut Microbiota in ApoE-/- Mice on a Western Diet"

_microorganisms, 2020, doi:10.3390/microorganisms8081225_

Round 1

Reviewer 1 Report

Dear Editor,

I carefully read the manuscript by Jiang et al., which is of outstanding interest and very well-written.

My comments for the authors:

  • the limitations of the study should be clearly addressed in the discussion
  • the oldest references should be replaced with newer articles
  • authors should refer to doi: 10.1007/s00394-020-02271-8.
  • authors should include a paragraph of "statistical methods".

Author Response

Dear reviewer,

Thank you very much for the opportunity to revise our manuscript. We appreciate the kind comments for the manuscript.

According to your advice, we amended the relevant part in manuscript.

Point 1: The limitations of the study should be clearly addressed in the discussion.

Response 1: The limitation of the study has been discussed in the line 517-521 of the manuscript and described as “However, the further preclinical studies are needed to reveal the mechanism of how L. mucosae A1 affects the metabolic endotoxemia and impaired lipid metabolism of host with more animal models with hyperlipidemia, and the efficiency of L. mucosae A1 on alleviating severe lipid accumulation and shifting gut microbiota on patients with dyslipidemia and atherosclerosis requires to evaluated in clinical study.”

Point 2: The oldest references should be replaced with newer articles.

Response 2: References no.1/2/3/4/15/16/55 have been replaced with newer articles.

1)Old reference 1:Weber, C.; Noels, H. Atherosclerosis: current pathogenesis and therapeutic options. Nature Medicine 2011, 17, 1410-1422, doi:10.1038/nm.2538.

New reference 1:Koliaki, C.; Liatis, S.; Kokkinos, A. Obesity and cardiovascular disease: revisiting an old relationship. Metabolism 2019, 92, 98-107, doi:10.1016/j.metabol.2018.10.011.

2)Old reference 2:Nelson, R.H. Hyperlipidemia as a Risk Factor for Cardiovascular Disease. Primary Care 2013, 40, 195-+, doi:10.1016/j.pop.2012.11.003.

New reference 2:Laufs, U.; Parhofer, K.G.; Ginsberg, H.N.; Hegele, R.A. Clinical review on triglycerides. Eur. Heart J. 2020, 41, 99-109, doi:10.1093/eurheartj/ehz785.

3)Old reference 3:Lopez-Miranda, J.; Williams, C.; Lairon, D. Dietary, physiological, genetic and pathological influences on postprandial lipid metabolism. British Journal of Nutrition 2007, 98, 458-473, doi:10.1017/s000711450774268x.

New reference 3:Mach, F.; Baigent, C.; Catapano, A.L.; Koskinas, K.C.; Casula, M.; Badimon, L.; Chapman, M.J.; De Backer, G.G.; Delgado, V.; Ference, B.A., et al. 2019 ESC/EAS Guidelines for the management of dyslipidaemias: lipid modification to reduce cardiovascular risk. Eur. Heart J. 2020, 41, 111-188, doi:10.1093/eurheartj/ehz455.

4)Old reference 4:Klop, B.; Elte, J.; Cabezas, M. Dyslipidemia in Obesity: Mechanisms and Potential Targets. Nutrients 2013, 5, 1218-1240, doi:10.3390/nu5041218.

New reference 4:Peng, J.; Luo, F.; Ruan, G.; Peng, R.; Li, X. Hypertriglyceridemia and atherosclerosis. Lipids Health Dis 2017, 16, 233:1-233:12, doi:10.1186/s12944-017-0625-0.

5)Old reference 15:Ryan, P.M.; Ross, R.P.; Fitzgerald, G.F.; Caplice, N.M.; Stanton, C. Functional food addressing heart health: do we have to target the gut microbiota? Current Opinion in Clinical Nutrition and Metabolic Care 2015, 18, 566-571, doi:10.1097/mco.0000000000000224.

New reference 15:He, M.; Shi, B. Gut microbiota as a potential target of metabolic syndrome: the role of probiotics and prebiotics. Cell Biosci 2017, 7, 54:`1-54:14, doi:10.1186/s13578-017-0183-1.

6)Old reference 16:He, M.; Shi, B. Gut microbiota as a potential target of metabolic syndrome: the role of probiotics and prebiotics. Cell & Bioscience 2017, 7, doi:10.1186/s13578-017-0183-1.

New reference 16:Cicero, A.F.G.; Fogacci, F.; Bove, M.; Giovannini, M.; Borghi, C. Impact of a short-term synbiotic supplementation on metabolic syndrome and systemic inflammation in elderly patients: a randomized placebo-controlled clinical trial. Eur J Nutr 2020, doi:10.1007/s00394-020-02271-8.

7)Old reference 55:Caesar, R.; Fak, F.; Backhed, F. Effects of gut microbiota on obesity and atherosclerosis via modulation of inflammation and lipid metabolism. Journal of Internal Medicine 2010, 268, 320-328, doi:10.1111/j.1365-2796.2010.02270.x.

New reference 55:Moreira, A.P.B.; Texeira, T.F.S.; Ferreira, A.B.; Peluzio, M.D.G.; Alfenas, R.D.G. Influence of a high-fat diet on gut microbiota, intestinal permeability and metabolic endotoxaemia. Br. J. Nutr. 2012, 108, 801-809, doi:10.1017/s0007114512001213.

Point 3: Authors should refer to doi: 10.1007/s00394-020-02271-8.

Response 3: We have referred to the article in the line 61 of the manuscript. The citation number of the reference is 16.

Point 4: Authors should include a paragraph of "statistical methods".

Response 4: In the previous manuscript, the statistical methods were described following each measurement in the method. According to your advice, we have set up a separate paragraph 2.9 for “Statistical Analysis for Animal Trial” in the line 247-251 of the present manuscript. And the statistical analysis for gut microbiota profiling is still in the separate paragraph 2.10 as it is quite different from the statistical analysis for animal trial.

Reviewer 2 Report

In this work, authors describe the isolation of a Lactobacillus mucosae strain, identified as A1, which was promoted by diet enriched with fibers in a patient with Prader-Willis syndrome. This strain was associated with reduction in circulating lipids and positive effects in mice with deleted ApoE gene. This is an interesting work, for which I have the following suggestions:

  • In the abstract, the aim is clearly expressed, but the experiments described in the work, as well as the findings are unclearly illustrated. Especially, the reference to the patient could be avoided.
  • Line 33, results.
  • Line 49, is instead of was.
  • Line 53, spell out or explain FMO3.
  • Line 57, is instead of are.
  • The paragraph 2.4 should be the first in Methods.
  • Line 155: describe how blood was obtained by the tail.
  • Line 158: how were the animals killed?
  • A separate paragraph describing the statistical analysis is required.
  • Line 282, delete undoubtedly, nothing is without doubts in science.
  • Add indication of scale bar measure in legend to figure 2.
  • In discussion, it is not mentioned that the authors findings’ could be important for patients maintained on the ketogenic diet for pharmacoresistant epilepsy, illustrated by Giordano et al. 2014 (doi: 10.3389/fneur.2014.00063), or for those with Autosomal Dominant Polycystic Kidney Disease, as described by Testa et al 2019 (10.1016/j.phanu.2019.100154) and 2020 (10.1016/j.phanu.2020.100206), which present increased levels of cholesterol as adverse event.

Author Response

Dear reviewer,

Thank you very much for the opportunity to revise our manuscript. We appreciate the kind comments for the manuscript. According to your advice, we amended the relevant part in manuscript.

Point 1: In the abstract, the aim is clearly expressed, but the experiments described in the work, as well as the findings are unclearly illustrated. Especially, the reference to the patient could be avoided.

Response 1: In the line 15-32, the abstract has been amended as “Supplementation of probiotics is a promising gut microbiota-targeted therapeutic method for hyperlipidemia and atherosclerosis. However, the selection of probiotic candidate strains is still empirical. Here, we obtained a human-derived strain, Lactobacillus mucosae A1, which was shown by metagenomic analysis to be promoted by high-fiber diet and associated with the amelioration of host hyperlipidemia, and validated its effect on treating hyperlipidemia and atherosclerosis as well as changing structure of gut microbiota in ApoE-/- mice on western diet. L. mucosae A1 attenuated the severe lipid accumulation in serum, liver and aortic sinus of ApoE-/- mice on western diet, while it also reduced the serum lipopolysaccharide-binding protein content of mice, reflecting the improved metabolic endotoxemia. In addition, L. mucosae A1 shifted the gut microbiota structure of ApoE-/- mice on western diet, including recovering a few members of gut microbiota enhanced by the western diet. This study not only suggests the potential of L. mucosae A1 to be a probiotic in the treatment of hyperlipidemia and atherosclerosis, but also highlights the advantage of such function-based rather than taxonomy-based strategy for selection of candidate strains for the next generation probiotics.”

Point 2: Line 33, results.

Response 2: We have changed “result” to “results” in the line 37.

Point 3: Line 49, is instead of was.

Response 3: We have changed “was” to “is” in the line 53.

Point 4: spell out or explain FMO3.

Response 4: The full name (flavin monooxygenase) and the function of FMO has already been described in the line 55. We have added an explanation for FMO3 as “the most active member in metabolizing TMA to TMAO of FMO family in the liver” in the line 57.

Point 5: Line 57, is instead of are.

Response 5: We have change “are” to “is” in the line 62.

Point 6: The paragraph 2.4 should be the first in Methods.

Response 6: The paragraph 2.4 is the method for animal trail design, in which L. mucosae A1 was needed. The method for the isolation and genomic characterization of L. mucosae A1 should be put forwards to the animal design. And in this way, the order of the methods can be consistent with the order of the result description.

Point 7: Line 155: describe how blood was obtained by the tail.

Response 7: We have added the description of tail bleeding in detail as “In detail, we secured mice in a restraint tube, wiped the tail with warm water to cause vasodilation, snipped the very end of the tail with a sterile number 11 scalpel, stroked the tail to collect blood into a collection tubes and stopped bleeding by applying pressure with a gauze pad as well as styptic powder to the tail tip.” in line 164-167.

Point 8: how were the animals killed?

Response 8: The mice were killed by cervical dislocation after the mice were anesthetized with isoflurance and exsanguinated by retro orbital bleeding. The description of this part has been added in the line 169-172 as “At the end of the experiment, all the mice were anesthetized with isoflurance after 6-hours fasting. Blood samples were collected by retro orbital bleeding. After exsanguination, mice were killed by cervical dislocation.”

Point 9: A separate paragraph describing the statistical analysis is required.

Response 9: In the previous manuscript, the statistical methods were described following each measurement in the method. According to your advice, we have set up a separate paragraph 2.9 for “Statistical Analysis for Animal Trial” in the line 247-251 of the present manuscript. And the statistical analysis for gut microbiota profiling is still in the separate paragraph 2.10 as it is quite different from the statistical analysis for animal trial.

Point 10: Line 282, delete undoubtedly, nothing is without doubts in science.

Response 10: We have changed “undoubtedly belonged to the species L. mucosae” to “A1 strain belonged to the species L. mucosae” in the line 307.

Point 11: Add indication of scale bar measure in legend to figure 2.

Response 11: In the previous figure 2, the scale bar of the photomicrographs is too small to be noticed. We have added an obvious scale bar presented as the black lines on lower right of the photomicrographs, and described it as “The scale bar is 200 μm” in legend to figure 2 in the line 360 and 364.

Point 12: In discussion, it is not mentioned that the authors findings’ could be important for patients maintained on the ketogenic diet for pharmacoresistant epilepsy, illustrated by Giordano et al. 2014 (doi: 10.3389/fneur.2014.00063), or for those with Autosomal Dominant Polycystic Kidne;y Disease, as described by Testa et al 2019 (10.1016/j.phanu.2019.100154) and 2020 (10.1016/j.phanu.2020.100206), which present increased levels of cholesterol as adverse event.

Response 12: We have mentioned the three references that reviewer provided for us in the discussion and described in the line 513-516 as “Upon the function of L. mucosae A1 on ameliorating lipid accumulation, the strain not only can be developed as a potential probiotic to treat hyperlipidemia and atherosclerosis, but may also can be used together with specific dietary intervention, such as ketogenic diet, to avoid side effect presented as elevated plasma lipid in treatment of other diseases”. The citation numbers of the references are 62-64.

Round 2

Reviewer 2 Report

All my comments have been addressed in an appropriate manner.